# Progressive Skeletal Muscle Atrophy in Muscular Dystrophies: A Role for Toll-Like Receptor-Signaling in Disease Pathogenesis

**DOI:** 10.3390/ijms21124440

**Published:** 2020-06-22

**Authors:** Boel De Paepe

**Affiliations:** Neuromuscular Reference Center, Ghent University Hospital, Corneel Heymanslaan 10, 9000 Ghent, Belgium

**Keywords:** muscular dystrophies, myositis, Toll-like receptors

## Abstract

Muscle atrophy is an active process controlled by specific transcriptional programs, in which muscle mass is lost by increased protein degradation and/or decreased protein synthesis. This review explores the involvement of Toll-like receptors (TLRs) in the muscle atrophy as it is observed in muscular dystrophies, disorders characterized by successive bouts of muscle fiber degeneration and regeneration in an attempt to repair contraction-induced damage. TLRs are defense receptors that detect infection and recognize self-molecules released from damaged cells. In muscular dystrophies, these receptors become over-active, and are firmly involved in the sustained chronic inflammation exhibited by the muscle tissue, via their induction of pro-inflammatory cytokine expression. Taming the exaggerated activation of TLR2/4 and TLR7/8/9, and their downstream effectors in particular, comes forward as a therapeutic strategy with potential to slow down disease progression.

## 1. Muscular Dystrophies

Muscular dystrophies are hereditary disorders caused by defects in a plethora of genes involved in muscle protein expression and organization. These diseases share clinical features of progressive loss of muscle strength and dystrophic pathological patterns that can be discerned in the skeletal muscle tissue [1]. The most common type is Duchenne muscular dystrophy (DMD), a severe X-linked disorder that affects 1 in 3500 boys born worldwide. Due to disruptive mutations in the *DMD* gene, its protein product dystrophin is absent from muscle fiber membranes. Contraction of the dystrophin-deficient myofibers produces severe damage, generating cycles of muscle fiber necrosis and regeneration that gradually loose restorative efficiency. The disease usually presents in early childhood as progressive muscle weakness, which causes loss of ambulation by early adolescence. In addition to the skeletal muscle, the disorder also involves cardiac and respiratory muscles [2]. Currently, over 3000 mutations have been associated with the disease, most commonly the deletion of one or multiple exons (60–70%), point mutations (26%), and exon duplications (10–15%). A milder form of muscular dystrophy associated with *DMD* defects is Becker muscular dystrophy, a condition in which mutations in the gene lead to production of truncated, yet partially functional dystrophin protein. The limb girdle muscular dystrophies (LGMDs), a heterogeneous group of diseases caused by different gene defects, represent the second most common dystrophy, and associate with predominant shoulder girdle and pelvic muscle weakness. LGMDs are classified into dominant (formerly LGMD1, now LGMD D) and recessive (formerly LGMD2, now LGMD R) conditions, and are described by the gene defect they are caused by. Facioscapulohumeral muscular dystrophy (FSHD) most often affects facial, shoulder blade, and upper arm muscles, with patients displaying prolonged muscle contractions. FSHD can be distinguished into 2 types, with FSHD1 being by far the most common form, accounting for 95 percent of cases. Autosomal dominant FSHD1 is caused by abnormal expression of *DUX4*; FSHD2 shows digenic inheritance with alterations in genes associated with chromatin repression *SMCHD1* and *DNMT3B*. Myotonic dystrophy is the most common adult onset muscular dystrophy and is inherited in an autosomal dominant pattern. The cause of myotonic dystrophy type 1, known as Steinert disease, are alterations to *DMPK*, while myotonic dystrophy type 2 results from defects in *CNBP* (also known as *ZNF9*). Although molecular medicine is in full development, muscular dystrophies today still classify as incurable [3]. Understanding of human disease mechanisms strongly benefits from the available animal models. Many of the experimental data that will be discussed in this review have been generated using the spontaneous occurring mouse model for muscular dystrophy called the mdx. This is the most studied disease model for DMD, yet the disease phenotype of murine mdx is milder than that of the human condition. In early life, mice develop active degeneration and regeneration of the skeletal muscle tissue which, in later life, continues at a slower pace. Muscle atrophy becomes conspicuous from age 12 months, and at 24 months muscle mass is on average 40% reduced, partly attributable to a reduced muscle fiber cross-sectional area (CSA) [4]. 

The inflammatory component of muscular dystrophy pathogenesis is of pivotal importance for tissue regeneration via delicately regulated elimination and replacement. Damaged muscle fibers are cleared by inflammatory cells to allow new fibers to take their place. The complex interplay of soluble factors and adhesion molecules tightly regulates these immune responses. Myofibers become active participants in the regulation and recruitment of inflammatory cells, by inducing the major histocompatibility complex I (MHC-I) on their sarcolemma and by producing regulatory cytokines, steering the inflammation forward. These capabilities of dystrophic muscle cells to act as immune-active regulators have also been confirmed through in vitro studies. Myoblasts from LGMD patients with *LMNA* defects, constitutively secrete high levels of IL-6 and IL-8 [5], an expression pattern which resembles closely that of damaged cells. Other muscle tissue constituents, including blood vessels and infiltrating inflammatory cells, participate in the immune cascades, of which soluble cytokines are key regulators. DMD patients’ sera display elevated levels of interleukin (IL)-1, IL-6, and tumor necrosis factor (TNF)-α [6], and certain chemokines including CC-motif chemokine ligand (CCL) 2, CCL14, CXC-motif chemokine ligand (CXCL) 12, and CXCL14 [7]. Macrophages and other cytotoxic immune cells clear the muscle tissue of necrotic fibers, and most abundantly express CXCL8, CCL2, and CCL5. CCL2 in particular is produced by and targets monocytes and T-cells, driving forward inflammation in a self-amplifying process [8]. Initially started to allow muscle tissue recovery, inflammation perpetuates past its desired time point, aggravating damage. The important role played by the immune system means that immunosuppression is, at this time, standard treatment for DMD. Glucocorticoids mitigate symptoms and add years to patient mobility, but come with important side effects and do not represent a true cure for the disease.

## 2. Muscle Atrophy in Health and Disease

Healthy skeletal muscle tissue needs to adapt to an individual’s needs and physical activity by adjusting properties, size, and numbers of muscle fibers. The most obvious adaptation of muscle mass is associated with exercise, but other lifestyle factors, such as smoking [9], may also be of influence. Muscle mass and performance adaptation are under the dynamic control of factors involved in the regulation of protein expression and turnover. On the anabolic side, the insulin/insulin-like growth factor 1-pathway is an important stimulus for muscle mass buildup. On the catabolic side, skeletal muscle may produce myostatin, a secreted growth differentiation factor which promotes muscle atrophy [10]. Muscle maintenance is achieved via clearance of redundant and damaged proteins by coordinated actions of the cell’s proteolytic systems termed the ubiquitin-proteasome pathway (UPP) and the autophagy lysosome machinery [11]. The UPP directs ubiquitin-tagged substrates to the 26S proteasome complex, and the muscle-specific E3 ubiquitin ligases atrogin-1 and Muscle ring finger-1 (MuRF-1) are major regulators activated in muscular atrophy. Via transient increases of atrogin-1 and MuRF-1 gene expression, a shift in protein balance from synthesis to degradation is accomplished, leading to the loss of muscle mass. The system condemns proteins to degradation in a highly organized fashion. Atrogin-1 regulates global protein synthesis inhibition via degradation of the muscle differentiation factors, myogenic determination factor 1 (MyoD) and eukaryotic translation initiation factor 3f (eIF3f). MuRF-1 regulates selective degradation of sarcomeric proteins, including troponins, myosins, titin, and myotilin [12]. The lysosome-mediated intracellular protein degradation via autophagy pathways participates in the turnover of damaged proteins and organelles. Muscle fiber size may vary by loss or gain of cell material, i.e., organelles, cytoplasm, and proteins [13]. Autophagic clearance of defective mitochondria conveniently termed mitophagy is essential to replace mitochondria in order to satisfy the high energy demand of skeletal muscle tissue. Such replacement of mitochondria is complexly regulated and is dependent on the demands of the tissues. Disuse-associated muscle atrophy is characterized by prominent reduction of mitochondrial volume density [14]. When muscle fibers are damaged, they can be replaced by stimulating satellite cells to proliferate, differentiate, and fuse, to form new functional muscle fibers. Efficient skeletal muscle tissue regeneration is ensured by the induction of myogenic regulatory factors such as myogenic factor 5 (Myf5), MyoD, myogenin, and myogenic regulatory factor 4 (MRF4), which orchestrate restorative cascades. This highly coordinated recovery mechanism closely resembles embryonic muscle development, engaging factors that transform satellite cells into fully differentiated muscle fibers. In these processes, an important class of additional regulatory molecules are the non-coding RNAs. Untranslated transcripts are involved in target gene transcription, splicing, and translation, and the modification of RNAs. Perturbed non-coding RNA levels have been implicated in muscle disease [15]. Many micro RNAs associate with tissue degeneration/regeneration, inflammation, cell differentiation, and fibrosis. The circulating levels of the muscle-specific micro RNAs, miR-23a, miR-29b, miR-206, and miR-499, correlate with muscle atrophy, and long non-coding RNAs with both atrophy-promoting (SMN-AS1, Pvt1) and atrophy-inhibiting (lncISR1, MUMA) activities have been recognized [16].

In muscular dystrophies, skeletal muscle function becomes compromised, hence cellular factors that are involved in injury repair and disease recovery become activated. Growth factors stimulate quiescent satellite cells to form new myotubes. In an attempt to restore muscle tissue function, inflammatory cells are activated to aid tissue remodeling. Multiple mediators that regulate further immune cell recruitment are released; however, the tissue does not succeed to fully regenerate. In muscular dystrophy, dysregulated expression of regulatory non-coding RNAs is involved in the failing muscle regeneration [17,18] and represents relevant biomarkers for the evaluation of disease progression and therapeutic effects. Certain miRs have repeatedly been reported elevated in muscular dystrophy patients, hence are termed dystromiRs. DMD biopsies show elevated levels of miR-1, miR-31, miR-133 and miR-206 [19,20] and long non-coding Lnc-31 [21]. The less severe dystrophy associated with Becker muscular dystrophy is represented by a more moderate increase of atrophy-associated miR-206, and miR-1 and miR-133, in comparison to DMD [22].

Muscular dystrophies are characterized by enhanced tissue damage, necessitating clearance of dysfunctional proteins, yet exaggerated proteasome activity compromises muscle recovery. In the mdx mouse, taming the overactive proteasome offers a valuable therapeutic approach [23]. Overexpression of UPP-associated genes is also observed in LGMDs. Calpainopathy patients demonstrate increased MuRF-1 expression [24,25], in patient with dysferlinopathy both atrogin-1 and MuRF-1 expression levels become increased [26]. Interestingly, a defect in the E3 ubiquitin ligase tripartite motif 32 (TRIM32) causes recessive LGMD (formerly termed LGMD2H, now LGMD R8 TRIM32-related). TRIM32 is not directly involved in atrophic processes, yet regulates regeneration of muscle in response to atrophy, via satellite cell proliferation [27]. In addition to enhanced proteasome activity, autophagic signaling appears dysfunctional in muscular dystrophies. Such dysfunction may, however, result from both insufficient as well as excessive autophagy. Insufficient autophagy is observed in DMD and the mdx mouse model [28], while increased autophagy is observed in a mouse model of congenital merosine-deficient muscular dystrophy [29]. Disease mechanisms are associated with the activation of processes ending in necrotic and apoptotic muscle cell death. Although ubiquitously expressed in muscle, calpains become increased in mdx [30]. DMD and FSHD patients display increased amounts of apoptotic myonuclei, and the number of B-cell lymphoma 2 (Bcl-2) and Bcl-2-associated X protein-positive myofibers increases. While not present in healthy skeletal muscles, caspases 2, 3, 5, 7, 8, and Granzyme B become expressed in DMD muscle [31].

Loss of muscle mass is a prominent feature in muscular dystrophies, but not all muscle groups are equally affected. Changes to the muscle volume and the extent of fatty replacement differ in rate and progression between muscles. DMD quadriceps develop muscle fiber atrophy that gradually worsens with age [32]. The CSA of the quadriceps femoris muscle is significantly smaller in DMD patients between 8 and 14 years old than in their healthy age-matched controls [33]. In the mouse model of DMD, CSA of diaphragm fibers becomes significantly smaller than that of age-matched controls when mdx mice reach the age of 9 months. In addition to such temporal accumulation of diaphragm pathology with advancing age, spatial variations are noted, pointing to non-uniform remodeling [34]. In contrast, DMD children under 12 years of age display sustained hypertrophy of the gastrocnemius as a compensation for surging muscle weakness. This phenomenon has also been observed in animal models of the disease. Upregulation of the myostatin-regulating miRNAs, miR-539 and miR-208b, could be a possible explanation for hypertrophy of cranial sartorius in the golden retriever muscular dystrophy model [35].

## 3. Toll-Like Receptor Signaling and Muscular Atrophy

Toll-like receptors (TLRs) represent essential components of host defense, detecting and neutralizing infections by activating the innate immune system. In addition, TLRs sense disrupted tissue integrity, reacting to danger-associated molecular patterns (DAMPs) released from damaged cells. Ten human (TLR1-10) and twelve murine (TLR1-9, TLR11-13) TLRs have been characterized. They are classified into extra-cellular and intra-cellular receptors based upon their location of expression. The effects of TLR stimulation and inhibition on muscle atrophy can be revealed in vitro in muscle cells and by investigating the phenotype of mouse models (Table 1). Most TLRs exhibit redundant responses, yet ligand-related effects and tissue-specific expression may produce different outcomes. TLR-mediated signaling is imperative for the elimination of infections and for promoting tissue repair. In order to protect, however, their activities need to be kept on a tight leash. Dysregulation of the interactions between TLRs and endogenous metabolites can lead to chronic and destructive inflammation.

### 3.1. Cell Surface TLRs

Receptors TLR1, TLR2, TLR4, TLR5, TLR6, TLR10, and murine TLR11 are expressed on the cell surface. TLR1 and TLR6 recognize microbial lipopeptides and derivatives originating from damaged cells. TLR2 and TLR4 bind a wide array of ligands, many of which they share with each other. They are essential receptors for bacterial lipopolysaccharide (LPS) but also recognize multiple endogenous ligands. For instance, tissue damage caused by inflammation results in the release of the TLR2/4 ligand high mobility group Box 1 (HMGB1) from the muscle tissue [42]. When secreted, the intracellular transcription factor becomes a soluble activator of pro-inflammatory cytokines [43]. In addition, the interaction of HMGB1 with TLR4 has been reported to stimulate the expression of MHC-I in murine muscle tissue and in peripheral blood mononuclear cells [44]. MHC-I enables muscle fibers to present antigen and become active participants in immune reactions. Similarly, heat shock proteins (HSPs) that function as intracellular molecular chaperones become TLR2/4-binding DAMPs once released from damaged cells. Other endogenous tissue damage signals that engage TLR2/4 are fibronectin domains, oligosaccharides of hyaluronic acid, heparan sulfate, and fibrinogen, which require high concentrations to activate the receptors. Acute-phase protein serum amyloid A1 (SAA1) is another ligand for TLR2/4 mostly expressed in the liver, but skeletal muscle appears to be an additional contributing source. SAA1 levels are influenced by pro-inflammatory cytokine expression, and high levels of SAA1 present in a mouse allograft model can be normalized by treatment with anti-IL6R antibody therapy [45]. TLR5 recognizes an evolutionarily conserved domain of flagellin, the major structural protein of bacterial flagella. TLR10 remains a receptor without clearly defined ligands, although it has been described to recognize human immunodeficiency virus proteins [46]. Mouse TLR11 binds protozoan profilin.

Expression of cell surface TLRs influences muscle function greatly. Stimulating TLR4 leads to the release of pro-inflammatory CCL2 from muscle cells in vitro [47]. Mice deficient for TLR2 exhibit a delayed clearance of necrotic tissue after muscle injury due to reduced numbers of infiltrating macrophages [48]. TLR2/4 ligand SAA1 is endogenously expressed in human myotubes in culture and further increases when cells are exposed to IL-6 and TNF-α [38]. SAA1 induces myotube atrophy in murine muscle cells in culture via its interaction with TLR2/4, and causes a 5-fold upregulation of its receptor TLR2 [36]. TLR2/4 activation by extracellular HSP70 and HSP90 provokes muscle catabolism by direct activation of the UPP [49]. TLR2/4 agonist LPS induces autophagy-related *Atg6,7,12* genes in muscle cells [50] and activation of UPP and autophagy [51]. In addition, LPS treatment induces expression of *MuRF1* [52] and *atrogin-1* [53], resulting in loss of muscle mass. In rats, an LPS challenge causes a significant decrease in gastrocnemius muscle fiber diameter, and a significant elevation in the muscle proteolysis rate [54]. In contrast, TLR4 knockout mice appear protected from such ligand-induced muscle catabolism. Lack of TLR4 function in mice does not alter gastrocnemius muscle mass and CSA [55], while absence of the receptor protected against HSP70 and HSP90 induced muscle wasting [56]. 

### 3.2. Intracellular TLRs

Intracellular TLR3, TLR7, TLR8, TLR9, and murine TLR12 and TLR13 are involved mostly in surveillance for viral infection. Their activation stimulates the production of antiviral cytokines, such as interferon (IFN) α/β, IL-1β, and IL-6, which suppress viral replication. These TLRs associate with intracellular endosomes, where they have access to viral nucleic acids. TLR3 recognizes retroviral double-stranded RNA genomes, while TLR7 and TLR8 bind single-stranded RNA. TLR9 localizes to the endoplasmic reticulum but, upon stimulation, traffics through the Golgi toward the endolysosomal compartment, where it is cleaved to its DNA-sensing form [57]. The receptor is activated by bacterial DNA sequences containing hypomethylated CpG motifs [58]. In addition to such infection surveillance, TLR9 can function as a sensor for cellular damage. Insults that lead to compromised mitochondrial function and oxidative stress through excessive reactive oxygen species (ROS) production generate mitochondrial debris [59], which behaves as DAMPs and activates recognition receptors [60]. Mitochondrial DNA closely resembles the genome of its bacterial ancestry, enabling recognition by TLR9. Release of mitochondrial DNA stems from cell necrosis but may also result from disturbed mitochondrial permeability via opening of the mitochondrial transition pore [61]. TLR9 activation triggers pro-inflammatory cytokine release [62], amplifying and sustaining inflammatory responses. Mouse TLR11 and TLR12 bind bacterial profilin, while TLR13 recognizes bacterial ribosomal RNA.

Intracellular TLRs are also important influencers of muscle function and are potent regulators of muscle fiber size and tissue inflammation, with both TLR3 and TLR9 involved in signaling. Intranasal administration of the exogeneous double-stranded RNA mimetic polyinosinic-polycytidylic acid reduces skeletal muscle mass and the CSA of gastrocnemius, soleus, and diaphragm fibers in mice, a process mediated via TLR3 binding [39]. Skeletal muscle that is defective for the mitochondrial fusion protein optic atrophy 1 exhibits mitochondrial DNA-mediated TLR9 activation, which leads to nuclear factor κB (NFκB) activation and subsequent myositis [63]. It has been observed that TLR9 signaling is directly linked with the mitophagic process, i.e., the autophagic removal of defective mitochondria, via interaction of the core autophagy protein beclin 1 with the receptor’s cytoplasmic domain [64]. The failing tissue recovery programs lead to gradual replacement of muscle fibers by extracellular matrix components. In muscular dystrophies, such fibrotic remodeling and the fatty tissue replacement that follows contribute to progressive muscle dysfunction [65]. TLR2/4 activation appears involved in fibrosis. In an experimental mouse peritoneal fibrosis model, deficiency protects completely (TLR2^-/-^) and partly (TLR4^-/-^) against fibrotic tissue remodeling [66].

### 3.3. TLR Effectors and Adaptor Molecules

TLRs are composed of three distinct domains: An extracellular domain that senses ligand, a transmembrane domain anchor, and an intracellular domain that translates activation into an appropriate cellular response. The cytoplasmic portion of TLRs shows high similarity to that of the IL-1 receptor family, and is therefore termed the Toll/IL-1 receptor (TIR) domain. Downstream cellular effects are generated through an association with various cytoplasmic adaptor proteins and the downstream signaling they can provoke. Myeloid differentiation primary response 88 (MyD88), TIR domain-containing adapter inducing interferon (TRIF), TRIF-related adapter molecule (TRAM), and TIR domain-containing adapter protein (TIRAP) mediate appropriate intracellular signaling cascades, activating differential sets of transcription factors. The different receptors use a specific combination of adaptor proteins, determining the resultant downstream signaling. TLR4 appears most versatile, as it can activate all four different downstream adaptors (Figure 1) [67]. TLR3 does not use the MyD88 dependent pathway, and is activated by TRIF-dependent signaling pathways only. TLR signaling cascades are complex and interconnected [68], and different adaptor molecules may result in a similar end-product. IFNα-production by plasmacytoid dendritic cells can be initiated by activation of TLR7 and TLR9, a process that is MyD88-dependent [69,70]. In contrast, stimulation of type 1 IFN-inducible genes initiated through TLR3 and TLR4 stimulation is MyD88-independent and involves TRIF-dependent pathways [69,70,71]. TLR cascades may involve IFN regulatory factors (IRFs), which regulate innate immunity via expression of type 1 IFNs and IFN-inducible genes. In addition, NFκB can be activated, which induces pro-inflammatory cytokine and chemokine gene expression. Another downstream effector is activator protein-1 (AP-1), which is involved in the regulation of the cell cycle, and in cell differentiation and survival [72]. 

## 4. Altered TLR Function in Muscular Dystrophies

Overactive TLR functions have been described in muscular dystrophies, implicating the receptors in the disease-associated tissue damage. Deficient muscle fibers undergo abundant necrosis, which releases signal molecules that can subsequently be captured by danger-sensing receptors, triggering exaggerated and sustained inflammatory responses. The increased activity of these pathways in muscular dystrophies could stem from perpetuated induction by DAMPs originating from damaged muscle and/or by failing feedback regulation mechanisms unable to deactivate the receptors, which result in higher levels of their executers. 

Firstly, increased receptor levels have been associated with muscular dystrophies. Human healthy control muscle has been typed TLR4, TLR7, and TLR9 immunonegative, while dystrophic muscle tissue gains TLRs [73]. Primary muscle cells from dysferlin-deficient mice express TLR2 and TLR4, and can efficiently produce IL-1β in response to LPS [74]. In muscle tissue from laminopathy patients, TLR4 is expressed on the sarcolemma of some muscle fibers and on blood vessels [73]. In mdx mice, *TLR4* mRNA expression in the diaphragm is 3-fold higher than in wild-type mice. This overexpression appears to aggravate disease, as *TLR4*-ablation improves diaphragm and tibialis anterior histopathological features and increases fiber size [75]. A possible underlying mechanism could be represented by the regulating activities of muscle-infiltrating macrophages. Induction of TLR4 favors pro-inflammatory processes, while TLR4 deficiency alters the phenotype of muscle-infiltrating macrophages toward tissue-restorative lineage cells. In muscle biopsies from DMD patients, TLR7 is upregulated at very early stages of the disease, localizing to inflammatory cells and blood vessels [76]. In muscle tissue from laminopathy patients, *TLR7* and *TLR9* mRNA are significantly increased, and many muscle tissue-infiltrating CD68+ macrophages are TLR7 and TLR9 positive. TLR7 is mostly expressed on blood vessels, while TLR9 localizes to the muscle fiber membrane, which occasionally also stains for TLR7 [73]. 

Secondly, dystrophic muscle displays increased levels of endogenous TLR ligands, which perpetuates the activation of the TLR pathway. The endogenous TLR2/4 ligand HSP70 is significantly elevated in DMD sera compared to age-matched controls, and levels are highest in young patients [77]. Similarly, HMGB1 is not detectable in the control muscle, but is present in the cytoplasm of mdx and DMD muscle fibers [75]. The TLR2/4/HMGB1 interaction initiates the production of MHC-I, TNF-α, and IL-6 in inflamed muscles [44]. Further, the TLR2/4 ligand fibrinogen links coagulation cascades with inflammation. mdx mice deficient in *Fib*, or treated with the fibrinogenolytic compound ancrod, show less severe fibrosis and muscle weakness, and reduced pro-inflammatory cytokine levels and macrophage infiltration [78].

Thirdly, perturbed expression of TLR downstream effectors has also been reported in muscular dystrophies. In DMD, increased expression of adaptor protein MyD88 has been described [79]. The question is raised whether this overexpression is a good or a bad thing for the regeneration of dystrophic muscle. On the negative side, satellite cell-specific deletion of MyD88 aggravates the disease phenotype of mdx. MyD88 deficiency leads to loss of skeletal muscle mass and increased fibrosis [80]. It seems that MyD88 activity is essential for satellite cells to uphold their proliferative capacities and to allow their fusion with injured muscle fibers. Nonetheless, in *DYSF*-*MyD88* double deficient mice, disease phenotype improves, with increases in body weight and muscle strength [81]. This protective effect is possibly due to diminished tissue inflammation.

It appears that sustained TLR pathway activation in the muscle tissue of muscular dystrophy has mostly negative effects, perpetuating tissue damage via muscle fiber loss and enhanced inflammation (Figure 2).

## 5. Toll-Like Receptors as a Therapeutic Target for Muscular Dystrophy 

Quenching overactive TLR activity to an appropriate level whilst avoiding detrimental consequences of complete receptor inhibition could represent a therapeutic route for muscular dystrophies. Based upon the activation of TLR pathways observed in animal models and in human studies, TLR2/4 and TLR7/8/9 pathways surface as plausible therapeutic targets to be further explored (Table 2). An important advantage of targeting TLRs is that these pathways participate in a wide spectrum of diseases, in which case drugs already in development for treating other disorders may be repurposed for muscular dystrophies. The regulation of the interactions between dystrophin deficiency and the perseverance of chronic inflammation is an amenable therapeutic target. To our advantage, TLR activity exhibits context-dependent roles, with disruption of their activities showing negative effects mostly in healthy muscle and less so in muscular dystrophy. When normal muscle lacks TLR2 function for instance, this hinders repair after acute tissue injury, but, on the contrary, blocking the receptor appears beneficial to muscle repair in DMD. Absence of TLR2 in mdx mice significantly reduces macrophage numbers and shifts their phenotype in favor of the anti-inflammatory M2 lineage [48]. A similar phenotypic shift toward the M2 macrophage phenotype is also present in TLR4-deficient mdx [75]. The phenotype of macrophages is crucial for efficient muscle tissue reconstitution. M1 macrophages are present in the inflammatory period and associate with phagocytosis; M2 macrophages accumulate at injury sites once necrotic tissue has been removed and participate in the regeneration, remodeling and restoration of the muscle tissue. If this succession is disturbed, the tissue loses restorative capacities.

A first, obvious therapeutic strategy could be to neutralize TLR activities with selective receptor antagonists. Many TLR-targeted compounds, both as stand-alone drugs and in combinatorial therapeutics as adjuvants, are currently tested in clinical trials at various stages. Small molecule therapeutics reactive to TLRs have been developed which exhibit anti-inflammatory properties, including TLR2 antagonist OPN-305; TLR3 antagonist PRV-300; TLR4 antagonists NI-0101, eritoran, MPL, JKB-121, and JKB-122; TLR9 antagonist hydrochloroquine; and compounds that antagonize multiple TLRs such as IMO-8400, CPG-52364, and VB-201. The TLR7/8/9 antagonist IMO-8503, for instance, is a potent inhibitor of cancer-induced cachexia [82]. Similar activities toward muscular dystrophy-associated muscle wasting might be expected, as the therapeutic potential of an oligonucleotide-based antagonist of TLR7/8/9 has been reported in the mdx model. The study shows that treatment of young dystrophic mice increases muscle strength and reduces skeletal muscle inflammation by decreasing the levels of inflammatory cytokines IFNγ and IL-1β. This effect could be confirmed in mouse and human cell-based assays, where the antagonist inhibits the expression of the pro-inflammatory cytokines TNF-α, IL-6, IL-12, IFN-α, IL-1β, and the chemokine CXCL10 [83]. Further evidence stems from experiments with the TLR7 agonist ORN06 which, on the contrary, is able to enhance the production of TNF-α and CCL2 in muscle cells [79]. 

A different strategy to alleviate dystrophy could be to diminish the levels of muscular dystrophy-associated DAMPs. TLR2/4 ligand HMGB1 is found in the cytoplasm of non-necrotic fibers from both humans and mice with muscular dystrophy, suggesting muscle cells actively secrete HMGB1. The large numbers of infiltrating macrophages are another potential source of soluble HMGB1. Treating mdx mice with glycyrrhizin, a compound which inhibits HMGB1 via binding with HMG boxes, improves diaphragm maximal force-generating capacity, and decreases levels of fibrosis and macrophage infiltration [75]. Diminishing HMGB1 levels is a potent immune-deactivating strategy. In peripheral blood mononuclear cells derived from mice with experimentally induced autoimmune myositis, treatment with anti-HMGB1 reduces their expression of MHC-I, TNF-α, and IL-6 [44].

Another strategy might be to reduce mitochondrial DAMP-induced TLR activation by administering antioxidants that support free-radical scavenging systems and prevent excessive ROS production [60]. In support, treatment of mdx mice with the cysteine precursor N-acetylcysteine prevents treadmill exercise-induced muscle fiber necrosis [84]. N-acetylcysteine-treated mdx mice also display improved grip strength, reduced hypertrophy of the extensor digitorum longus, increased mitochondrial function, and less severe inflammation in gastrocnemius muscle [85]. In addition, N-acetylcysteine supplementation rescues mdx diaphragm function, reducing fibrosis and inflammatory cell infiltration [86]. Though treated mdx mice display increased force, N-acetylcysteine supplements come with the important side effects of reduced muscle weight and impaired body weight gain in growing mice [87]. Unfortunately, placebo-controlled clinical trials evaluating antioxidants in DMD have been mostly negative [88], but other TLR-dampening supplements might be attempted as safe supportive treatment. In rats, leucine supplementation alleviates LPS-induced skeletal muscle wasting and attenuates the decrease in gastrocnemius muscle fiber diameter via attenuated NF-κB mRNA expression. In an in vitro setting, leucine-induced decreases of muscle proteolysis and NF-κB activity is confirmed [54]. The data above firmly suggest that leucine supplementation could inhibit excessive skeletal muscle degradation, as well as enhance protein synthesis, attenuating the negative effects caused by LPS-TLR2/4 stimulation. In rohu fish, this protective effect of leucine has also been described both in vivo and in vitro. Leucine pre-supplementation could protect fish against LPS-induced inflammation through deactivation of the TLR4−MyD88 signaling pathway. Pretreatment of primary hepatocytes attenuates the excessive activation of LPS-induced TLR4 signaling as manifested by lower levels of TLR4, MyD88, MAPKp38, NF-κBp65, and increased levels of inhibitor κB (IκB) -α protein [89]. Nonetheless, an early study describes the failed therapeutic response of leucine dosed at 0.2 g per kg per day over a 1-year double-blind controlled trial in 96 DMD patients [90]. Glutamine, another amino acid supplement, has also been observed to influence TLR activity. Glutamine supplementation associates with down-regulation of *TLR4*, *MyD88*, and *TRAF6* expression, decreasing LPS-induced injury to mucosal cells [91]. Yet again, glutamine did not improve muscle function in a double-blind, randomized crossover trial conducted in 30 DMD patients [92].

Blocking downstream effectors of TLR signaling can be achieved by the small molecule inhibitor TAK-242, which selectively binds the intracellular domain of TLR4. Binding disrupts interaction of TLR4 with its adaptor molecules TIRAP and TRAM, preventing them to activate NFκB and IRFs [93]. TAK-242 has been observed to prevent tissue fibrosis in mouse models, reducing myofibroblast transdifferentiation [94]. TAK-242 has not been tested yet in muscular dystrophies, but in a mouse model of muscle disuse, the hind limb suspension-induced decline in soleus muscle mass could not be prevented by administering TAK-242 [41]. Targeting downstream NFκB directly would, of course, also alleviate inflammation, an approach proposed for combating inflammatory disease of various etiology [95]. NFkB pathway IκB kinase inhibitor Bristol-Myers Squibb (BMS)-345541 diminishes inflammation-induced atrophy in the gastrocnemius and plantaris of a mouse model (Hahn et al. [36]), but is ineffective in reducing overactive NFκB in adult mdx muscles [96]. NF-κB pathway inhibition by overexpression of de-ubiquitination protein A20, is, however, able to attenuate muscle inflammation and increase Myf5 levels in quadriceps, decreasing dystrophic pathology and improving muscle health in mdx mice [97].

## 6. Conclusions

The involvement of Toll-like receptor-signaling in muscle physiology is complex, as these receptors act as double-edged swords exhibiting context-dependent capabilities of promoting or inhibiting tissue growth and recuperation. Initially aimed at restoring harmed muscle fibers, the sustained injury-inflicted release of cellular content into the extracellular space in dystrophic muscle mediates chronic activation of the innate immune system, which leads to a perpetuated pro-inflammatory state. Activation via TLRs intended to optimize tissue recovery ends up overzealous, and aggravates tissue damage. Therapeutic compounds able to restrain and fine-tune immune responses by targeting the TLRs selectively and proportionately are, therefore, an amenable approach in the context of muscular dystrophies, the idea being to restore TLR pathways to normal surveillance mode and regain muscle tissue homeostasis. Better focusing immune reactions through administering drugs that selectively antagonize TLR function appears a valuable non-dystrophin-centered supportive treatment approach, and TLR2/4 and TLR7/8/9 interactions come forward as promising targets to be investigated further.

## Figures and Tables

**Figure 1 ijms-21-04440-f001:**
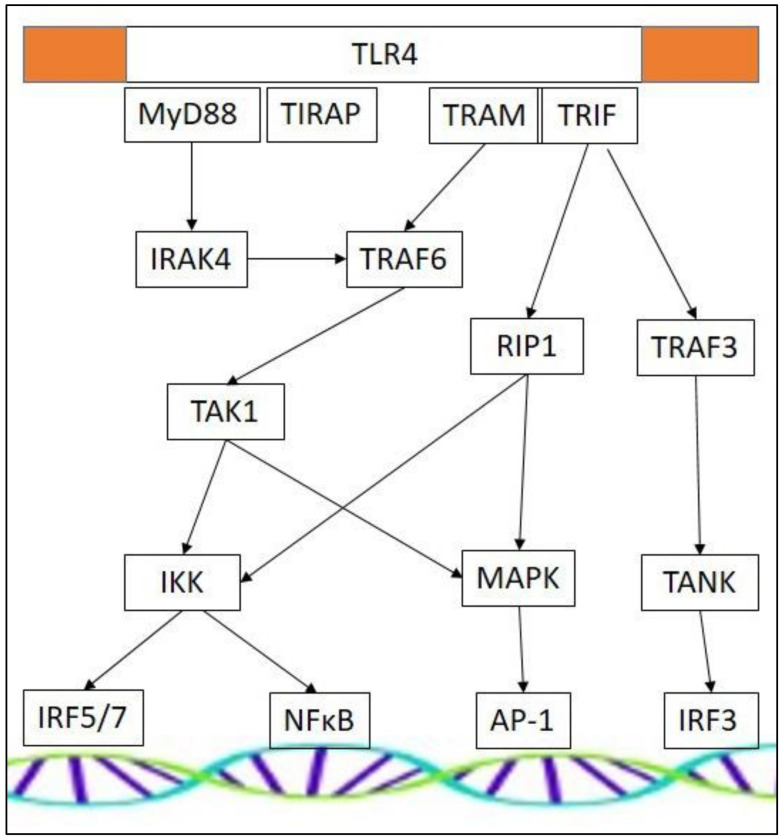
Signaling routes downstream of TLR4. Abbreviations: Activator protein-1 (AP-1), inhibitor of κB kinase (IKK), interleukin-1 receptor-associated kinase (IRAK), interferon regulatory factor (IRF), mitogen activated protein kinase (MAPK), myeloid differentiation primary response 88 (MyD88), nuclear factor κB (NFκB), receptor-interacting protein (RIP), Transforming growth factor-β-activating kinase (TAK), tumor necrosis factor receptor-associated factor (TANK), Toll/IL-1 receptor domain-containing adaptor protein (TIRAP), Toll-like receptor (TLR), tumor necrosis factor receptor-associated factor (TRAF), Toll/IL-1 receptor domain-containing adapter protein inducing interferon-β-related adapter molecule (TRAM), Toll/IL-1 receptor domain-containing adapter protein inducing interferon-β (TRIF).

**Figure 2 ijms-21-04440-f002:**
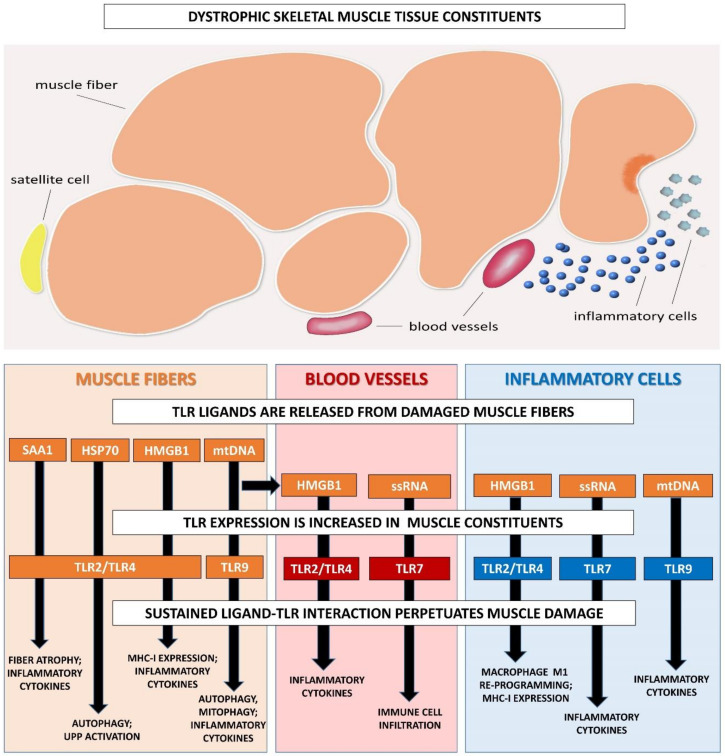
Model of TLR/ligand-driven tissue damage in muscular dystrophies. Proposed mechanism of perpetuated muscle damage in muscular dystrophies: Muscle contractions induce sustained muscle fiber injury that becomes chronic and can no longer be repaired. Due to unsealing of the muscle fiber membranes, intracellular acute-phase protein serum amyloid A1 (SAA1), heat shock protein 70 (HSP70), and high mobility group Box 1 (HMGB1) are released, and become Toll-like receptor (TLR) pathway-activating danger-associated molecular patterns (DAMPs). The corresponding cell surface receptors TLR2 and TLR4 are overexpressed in dystrophic muscle fibers, blood vessels, and tissue-infiltrating inflammatory cells which, upon stimulation, activate proteolytic and inflammatory processes, amplifying tissue damage further. Necrotic muscle fibers release nucleic acids that activate the corresponding cytoplasmic receptors. Released single-stranded RNA (ssRNA) sequences are picked up by blood vessels and inflammatory cells and bind to their receptor TLR7, activating degeneration/regeneration-focused immune cell function. Extracellular mitochondrial DNA (mtDNA) is engulfed by muscle fibers and inflammatory cells that possess its receptor TLR9, which subsequently leads to activation of autophagic processes and stimulates inflammatory responses even further.

**Table 1 ijms-21-04440-t001:** Effects of Toll-like receptor (TLR) pathway manipulation on muscle atrophy in vitro and in vivo.

Cell Line/Animal Strain	Treatment	Atrophic Effect	Reference
C2C12 myotubes	SAA1	Myotube atrophy and increased *IL-6* expression.	[36]
C2C12 myotubes	SAA1 + anti-TLR2 anti-TLR4	Attenuated myotube atrophy and IL-6 expression.	[36]
C2C12 myotubes	Pam3Csk4	Myotube atrophy.	[36]
C2C12 myotubes	LPS	Muscle cell atrophy.	[37]
C2C12 myotubes	LPS + TAK242	Attenuated muscle cell atrophy.	[37]
C2C12 myotubes	LPS	Myotube atrophy and increased SAA1 mRNA and protein.	[38]
C57BL/6J mice	Cecal ligation and puncture sepsis model	Increased expression of *TLR2* and *TLR4* in tibialis anterior and gastrocnemius/plantaris.	[36]
C57BL/6J mice	Poly(I:C)	Atrophy of gastrocnemius, tibilalis anterior, soleus, and plantaris, but not in extensor digitorum longus. Lower mean muscle fiber cross-sectional areas in gastrocnemius, plantaris, and diaphragm.	[39]
C57BL/6J mice	Mechanical ventilation + LPS	Diaphragmic atrophy.	[40]
C57BL/6J Tlr4Lps-del TLR4-deficient mice	Mechanical ventilation + LPS	Normal diaphragmic muscle fiber diameter.	[40]
C57BL/6J mice	Hindlimb suspension + TAK242	Smaller gastrocnemius, but muscle fiber size unaltered.	[41]

In the studies mentioned, cells and mice were treated with (1) TLR2/4 ligands: Acute-phase protein serum amyloid A1 (SAA1), lipopolysaccharide (LPS), Pam3Csk4; TLR3 agonist synthetic double stranded poly(I:C); TLR4 inhibitor TAK242.

**Table 2 ijms-21-04440-t002:** TLR pathway alterations in muscular dystrophy.

Gene Deficiency	Target	Regulation in Dystrophy	Reference
*DMD*	TLR2/4	Expression of TLR2 and TLR4 is significantly increased in DMD muscle.	[79]
		TLR4 expression is increased in mdx diaphragm.	[75]
		HMGB1 is overexpressed in the muscle fiber cytoplasm of DMD tissues.	[75]
		Soluble HSP70 is increased in serum of DMD patients.	[77]
	TLR7/8/9	TLR7 is expressed in inflammatory cells and blood vessels in muscle from DMD patients.	[76]
		Strong TLR7 staining is present in muscle fibers and inflammatory cells of DMD tissue.	[79]
		The vast majority of muscle cells isolated from mdx express TLR9.	[79]
	adaptors	MyD88 is upregulated in skeletal muscle of DMD patients.	[76]
		MyD88 protein levels are strongly increased in mdx muscle.	[80]
		MyD88 staining is increased in satellite cells of mdx mice.	[80]
*DYSF*	TLR2/4	Primary muscle cells prepared from SJL/J mice express TLR2 and TLR4.	[74]
	TLR7/8/9	TLR7 expression is significantly upregulated in muscle from A/J mice.	[81]
		TLR8 gene expression is not significantly altered in A/J mice.	[81]
*LMNA*	TLR2/4	TLR4 is strongly expressed on the sarcolemma of a subset of muscle fibers and capillaries in patient biopsies.	[73]
	TLR7/8/9	TLR7 is expressed in the endomysial space, on blood vessels, a minority of inflammatory cells, and occasionally at the sarcolemma of degenerated muscle fibers in patient biopsies.	[73]
		In patient muscle, TLR9 is expressed in the endomysial space, at the muscle fiber membrane, on capillaries, and rare inflammatory cells.	[73]

Abbreviations: High mobility group Box 1 (HMGB1), heat shock protein 70 (HSP70), myeloid differentiation primary response 88 (MyD88), Toll-like receptor (TLR).

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
