# Peer review of "Progressive Skeletal Muscle Atrophy in Muscular Dystrophies: A Role for Toll-Like Receptor-Signaling in Disease Pathogenesis"

_ijms, 2020, doi:10.3390/ijms21124440_

Round 1

Reviewer 1 Report

This review paper is well researched and the topic is original and timely.  It contains a large amount of background information about muscular dystrophies and TLRs.  However, there is relatively little information provided on the intersection of specific cellular mechanisms underlying muscle atrophy and how TLRs affect these pathways, which is surprising given the title of the review.  There is also ambiguity in discussing effects on atrophy versus other manifestations of muscle disease such as necrosis, fibrosis and muscle regeneration.  I believe this review has the potential to be a valuable addition to the literature with the following modifications:

Major:

1.    The review discusses other manifestations of muscle disease besides atrophy.  Therefore, I suggest a change in the title to reflect the broader pathology.  For example,  “A role for TLR signaling in muscular dystrophy pathogenesis”

2.      Remove Table 1 which is not directly contributory to the topic and can be found in any Neurology textbook.  It would be much more useful to have a figure showing different sites at which TLRs could act on disease pathogenesis (inflammatory cells, muscle fibers, satellite cells, blood vessels) and their downstream consequences of necrosis, fibrosis, and atrophy. This figure could also show the potential sources of endogenous TLR ligands such as HMGB1, heat shock proteins (HSPs) and serum amyloid A1 (SAA1).

3.      In addition, a table showing the different TLRs with their specific endogenous ligands and expression sites in different form of muscular dystrophy would fit well with the stated objective of the review.

4.      In general, the review is often too focused on TLRs in a general way and not sufficiently detailed regarding the specific cellular mechanisms that link TLRs to muscular dystrophy mechanisms. In other words, although the review explains in depth about the TLR signaling pathways, the linkage between these signaling pathways and muscular dystrophy manifestations is often not well articulated.

5.      For example, Figure 1 shows general TLR signaling components but without any linkage to the well known important muscle atrophy pathways. Specific cellular mediators of atrophy such as Atrogin-1, MuRF-1, Calpain, and Caspase-3 are not discussed.  A similar comment can be made about the effector proteins involved in autophagy and especially mitophagy, which require more discussion of their modulation by specific TLRs. 

6.      In addition to atrophy, what about links between TLR signaling and fibrosis pathways or the myogenic regulatory factors involved in regeneration?  Do TLRs alter the expression of these proteins or their function?

7.      The authors should discuss in more detail the question of where exactly the TLRs are exerting their adverse effects (which cell type: macrophages, lymphocytes, mature fibers, satellite cells?).  Also, does the stage of disease (early versus late) have any importance in the role of different TLRs? 

8.      In terms of future therapy, would it be best to inhibit TLRs on immune cells but try to preserve TLR function on the muscle fibers or satellite cells?  Is this even possible?  Are the immune cells more sensitive to TLR inhibitor drugs than the muscle fibers?  This could also be part of a perspective on future directions that need to be addressed in order to move the field forward.

Minor comments:

1.      There are some errors of English grammar and spelling throughout the paper (for example, on line 364 the word “instated” does not make sense).

2.      There are many statements that require addition of a reference (for example, the sentences which terminate at the end of line 299 and the beginning of line 309).  In addition, on lines 239-241, the author cited a reference 54 claiming “Human healthy control muscle is TLR4, TLR7 and TLR9 immunonegative, while DMD muscle tissue gains these TLRs.” However, this study does not include DMD muscle tissue.

3.      In lines 115 to 122, the mention of mdx mice as a model of human DMD seems out of place. It is suggested to introduce the topic of animal models and their limitations in section 1.

Author Response

Dear Guest Editors,

Dear Reviewers,

My sincerest thanks to the reviewers for their remarks regarding manuscript ijms-822835. The criticisms were pertinent, and I graciously admit that implementing the proposed changes has allowed considerable improvement of the review paper, both in content and in form. I address the issues raised point-by-point, and re-submit a new version of the manuscript, hoping it reaches the desired standards to merit further consideration for publication.

Reviewer 1

This review paper is well researched and the topic is original and timely.  It contains a large amount of background information about muscular dystrophies and TLRs.  However, there is relatively little information provided on the intersection of specific cellular mechanisms underlying muscle atrophy and how TLRs affect these pathways, which is surprising given the title of the review.  There is also ambiguity in discussing effects on atrophy versus other manifestations of muscle disease such as necrosis, fibrosis and muscle regeneration.  I believe this review has the potential to be a valuable addition to the literature with the following modifications:

Major:

  1. The review discusses other manifestations of muscle disease besides atrophy. Therefore, I suggest a change in the title to reflect the broader pathology.  For example,  “A role for TLR signaling in muscular dystrophy pathogenesis”

The title has been changed to better reflect the complex involvement of TLR pathways in muscular dystrophy disease pathogenesis, which indeed includes but is far from limited to atrophy.

  1. Remove Table 1 which is not directly contributory to the topic and can be found in any Neurology textbook. It would be much more useful to have a figure showing different sites at which TLRs could act on disease pathogenesis (inflammatory cells, muscle fibers, satellite cells, blood vessels) and their downstream consequences of necrosis, fibrosis, and atrophy. This figure could also show the potential sources of endogenous TLR ligands such as HMGB1, heat shock proteins (HSPs) and serum amyloid A1 (SAA1).

As requested, former Table 1 has been eliminated. Indeed, it contained only standard information on muscular dystrophy subtypes which can easily be found in medical textbooks. Based upon the remarks made by reviewer 2, a short summary of disease classification is now contained in the first paragraph of section 1. To further accommodate focus of the special issue, a new Table 1 is now introduced, summarizing the effects of TLR pathway-manipulation on muscle atrophy. This addition will aid the reader to measure up the involvement of the pathway and to better find her/his way around the published literature.  To illustrate the sources of TLR ligands and localize the responsive tissue constituents, a new Figure 2 has been added. The figure offers a model of how TLR activation may sustain muscle damage. Inflammatory reactions, which can be beneficial as well as detrimental to tissue regeneration, appear to be one of the key tools in deciding the focus, extent and nature of tissue responses. Section 6 (conclusion) has been rewritten to clarify the modus operandi of normalizing TLR activity as a therapeutic approach.

  1. In addition, a table showing the different TLRs with their specific endogenous ligands and expression sites in different form of muscular dystrophy would fit well with the stated objective of the review.

Both reviewer 2 and myself agree with reviewer 1 that the paper would gain clarity by introducing such a table. The manuscript now includes the new Table 2 describing TLR pathway alterations in muscular dystrophies.

  1. In general, the review is often too focused on TLRs in a general way and not sufficiently detailed regarding the specific cellular mechanisms that link TLRs to muscular dystrophy mechanisms. In other words, although the review explains in depth about the TLR signaling pathways, the linkage between these signaling pathways and muscular dystrophy manifestations is often not well articulated.

Indeed, TLR signalling is involved in much more than muscle fiber atrophy. These receptors participate in many processes associated with muscular dystrophy pathogenesis.  The text has been reworked thoroughly to clarify links with muscle re/degeneration, autophagy, cell death, inflammation and fibrosis.

  1. For example, Figure 1 shows general TLR signaling components but without any linkage to the well known important muscle atrophy pathways. Specific cellular mediators of atrophy such as Atrogin-1, MuRF-1, Calpain, and Caspase-3 are not discussed. A similar comment can be made about the effector proteins involved in autophagy and especially mitophagy, which require more discussion of their modulation by specific TLRs.

My sincerest apologies for this oversight. Description of atrophy mediators and mitophagy should, of course, be part of this paper, and this information has been added to section 2.

  1. In addition to atrophy, what about links between TLR signaling and fibrosis pathways or the myogenic regulatory factors involved in regeneration? Do TLRs alter the expression of these proteins or their function?

This is an important point raised. Indeed, processes of muscle damage, injury repair and regeneration and, when that fails, replacement by fibrotic and fatty tissue, are all connected. Fibrosis did not receive due attention in the first version of the manuscript, and section 3.2. now includes discussion on interaction with fibrotic processes.

  1. The authors should discuss in more detail the question of where exactly the TLRs are exerting their adverse effects (which cell type: macrophages, lymphocytes, mature fibers, satellite cells?). Also, does the stage of disease (early versus late) have any importance in the role of different TLRs?

To improve overview and to aid discussion, the manuscript now contains the new Figure 2 illustrating how an overactive TLR pathway within the different constituents of muscle tissue could drive muscular dystrophy-associated muscle damage forward. At this time, the bulk of research regarding variations of TLR pathway involvement varying with age remains too small to allow founded prognoses of age- or disease stage-dependent responses to TLR inhibition. In this respect, results to be obtained in the mdx model will no doubt proliferate in the near future.

  1. In terms of future therapy, would it be best to inhibit TLRs on immune cells but try to preserve TLR function on the muscle fibers or satellite cells? Is this even possible?  Are the immune cells more sensitive to TLR inhibitor drugs than the muscle fibers?  This could also be part of a perspective on future directions that need to be addressed in order to move the field forward.

Unfortunately, this issue cannot (yet) be answered satisfactory. It is indeed known that TLR signalling is involved in proper muscle regeneration and is important for the functioning of satellite cells. However, in case of muscular dystrophy, the plot seems to thicken considerably. Section 4 touches on the essential role played by MyD88 in this respect. General ablation of MyD88, as could be expected, aggravates mdx, however appears to benefit dysferlin deficient mice, which is more unexpected. This observation still requires a substantiated explanation.

Minor comments:

  1. There are some errors of English grammar and spelling throughout the paper (for example, on line 364 the word “instated” does not make sense).

The text has been revised throughout, and ‘instated’ has been replaced by ‘intended’.

  1. There are many statements that require addition of a reference (for example, the sentences which terminate at the end of line 299 and the beginning of line 309). In addition, on lines 239-241, the author cited a reference 54 claiming “Human healthy control muscle is TLR4, TLR7 and TLR9 immunonegative, while DMD muscle tissue gains these TLRs.” However, this study does not include DMD muscle tissue.

Incorrect phrasing and erroneous referral to the publication of Cappelletti et al. has been remedied.

  1. In lines 115 to 122, the mention of mdx mice as a model of human DMD seems out of place. It is suggested to introduce the topic of animal models and their limitations in section 1.

Indeed, placing of this entry was not logical, and it has been moved to the first paragraph of section 1 (introduction).

Reviewer 2 Report

The paper from Boel de Paepe make a comprehensive review of the role for TOLL-like receptor-signaling in disease-associated muscle fiber atrophy. The molecular part is solid clear and well explained. Unfortunately the disease associated mechanism are less clear and need to be improved. I suggest to modify the introductive part with a more detailed description of the different muscular dystrophies. Morevoer table 1 should be reformatted giving more de tails and  respecting the up-to-date classification for every group of disorders. For example Limg Girdle Muscular Dystrophies (LGMD) are now classified into dominantand recessive and they are called LGMD D 1 to X and LGMD R 1 to X. Facioscapulohumeral dystrophy exists in two forms FSHD1 and FSHD2. Distal muscular dystrophies do not exists. We talk of distal myopathies. There exists two mytonic dystrophies type 1 (Steinert) and type 2.

Moreover I suggest to make a table reporting on the specific alteration of Toll-like receptor signaling in different muscular dystrophies with the reference of the paper reporting such alteration. In the same table should be reported the potential therapeutic target. 

Author Response

Dear Guest Editors,

Dear Reviewers,

My sincerest thanks to the reviewers for their remarks regarding manuscript ijms-822835. The criticisms were pertinent, and I graciously admit that implementing the proposed changes has allowed considerable improvement of the review paper, both in content and in form. I address the issues raised point-by-point, and re-submit a new version of the manuscript, hoping it reaches the desired standards to merit further consideration for publication.

Reviewer 2

The paper from Boel de Paepe make a comprehensive review of the role for TOLL-like receptor-signaling in disease-associated muscle fiber atrophy. The molecular part is solid clear and well explained. Unfortunately the disease associated mechanism are less clear and need to be improved. I suggest to modify the introductive part with a more detailed description of the different muscular dystrophies. Morevoer table 1 should be reformatted giving more details and  respecting the up-to-date classification for every group of disorders. For example Limg Girdle Muscular Dystrophies (LGMD) are now classified into dominantand recessive and they are called LGMD D 1 to X and LGMD R 1 to X. Facioscapulohumeral dystrophy exists in two forms FSHD1 and FSHD2. Distal muscular dystrophies do not exists. We talk of distal myopathies. There exists two mytonic dystrophies type 1 (Steinert) and type 2.

Shortcomings of the introduction section were also remarked by reviewer 1. Combining the criticisms of both reviewers, I opted to remove table 1, and replace it with the necessary background information on muscular dystrophy subtypes as requested by reviewer 2, contained as part of section 1 paragraph 1. The requested corrections put forward here have been made, as well as mention of the new terminology for LGMDs.

Moreover I suggest to make a table reporting on the specific alteration of Toll-like receptor signaling in different muscular dystrophies with the reference of the paper reporting such alteration. In the same table should be reported the potential therapeutic target.

Thank you for this excellent suggestion, that was also similarly proposed by reviewer 1. A new Table 2 has been added, describing the alterations to TLR pathways that have been reported in muscular dystrophy patients and disease models. This illustrates better how TLR2/4 and TLR7/8/9 pathways have come forward as most plausible targets to investigate for therapeutic purposes.
